# G ATA2 mediates the negative regulation of the prepro-thyrotropin-releasing hormone gene by liganded T3 receptor β2 in the rat hypothalamic paraventricular nucleus

Go Kuroda[1], Shigekazu Sasaki[1]*, Akio Matsushita[1], Kenji Ohba[2], Yuki Sakai[1], Shinsuke Shinkai[1], Hiroko Misawa Nakamura[1], Satoru Yamagishi[3], Kohji Sato[3], Naoko Hirahara[4], Yutaka Oki[5], Masahiko Ito[6], Tetsuro Suzuki[6], Takafumi Suda[1]

1 Second Division, Department of Internal Medicine, Hamamatsu University School of Medicine, Hamamatsu, Shizuoka, Japan, 2 Medical Education Center, Hamamatsu University School of Medicine, Hamamatsu, Shizuoka, Japan, 3 Department of Organ and Tissue Anatomy, Hamamatsu University School of Medicine, Hamamatsu, Hamamatsu, Shizuoka, Japan, 4 Division of Endocrinology and Metabolism, Department of Internal Medicine, Japanese Red Cross Shizuoka Hospital, Shizuoka, Shizuoka, Japan, 5 Department of Internal medicine, Hamamatsu Kita Hospital, Hamamatsu, Shizuoka, Japan, 6 Department of Virology and Parasitology, Hamamatsu University School of Medicine, Hamamatsu, Shizuoka, Japan

* sasakis@hama-med.ac.jp

**Data Availability Statement:** All relevant data are within the paper and its Supporting information files.

## Abstract

Thyroid hormone (T3) inhibits thyrotropin-releasing hormone (TRH) synthesis in the hypothalamic paraventricular nucleus (PVN). Although the T3 receptor (TR) β2 is known to mediate the negative regulation of the prepro-TRH gene, its molecular mechanism remains unknown. Our previous studies on the T3-dependent negative regulation of the thyrotropin β subunit (TSHβ) gene suggest that there is a tethering mechanism, whereby liganded TRβ2 interferes with the function of the transcription factor, GATA2, a critical activator of the TSHβ gene. Interestingly, the transcription factors Sim1 and Arnt2, the determinants of PVN differentiation in the hypothalamus, are reported to induce expression of TRβ2 and GATA2 in cultured neuronal cells. Here, we confirmed the expression of the GATA2 protein in the TRH neuron of the rat PVN using immunohistochemistry with an anti-GATA2 antibody. According to an experimental study from transgenic mice, a region of the rat prepro-TRH promoter from nt. -547 to nt. +84 was able to mediate its expression in the PVN. We constructed a chloramphenicol acetyltransferase (CAT) reporter gene containing this promoter sequence (rTRH(547)-CAT) and showed that GATA2 activated the promoter in monkey kidney-derived CV1 cells. Deletion and mutation analyses identified a functional GATA-responsive element (GATA-RE) between nt. -357 and nt. -352. When TRβ2 was co-expressed, T3 reduced GATA2-dependent promoter activity to approximately 30%. Unexpectedly, T3-dependent negative regulation was maintained after mutation of the reported negative T3-responsive element, site 4. T3 also inhibited the GATA2-dependent transcription enhanced by cAMP agonist, 8-bromo-cAMP. A rat thyroid medullary carcinoma cell line, CA77, is known to express the preproTRH mRNA. Using a chromatin immunoprecipitation assay with this cell line where GATA2 expression plasmid was transfected, we observed the recognition of the GATA-RE by GATA2. We also confirmed GATA2 binding using gel shift

**Funding:** This work was supported in part by a Grant-in Aid for Scientific Research to S.S. from the Ministry of Education, Culture, Sports, Science and Technology of Japan (grant number 15K09425, URL: https://nrid.nii.ac.jp/ja/nrid/1000020303547/). The funder had no role in study design, data collection and analysis, decision to publish, or preparation of the manuscript.

**Competing interests:** The authors have declared that no competing interests exist.

assay with the probe for the GATA-RE. In CA77 cells, the activity of rTRH(547)-CAT was potentiated by overexpression of GATA2, and it was inhibited in a T3-dependent manner. These results suggest that GATA2 transactivates the rat prepro-TRH gene and that liganded TRβ2 interferes with this activation via a tethering mechanism as in the case of the TSHβ gene.

## Introduction

The hypothalamus-pituitary-thyroid (H-P-T) axis plays the central role for maintaining thyroid hormone homeostasis [1–4]. Signals following food intake and cold exposure modulate the level of thyrotropin-releasing hormone (TRH) production in the hypothalamic paraventricular nucleus (PVN), resulting in fine-tuning of the regulation of thyrotropin (thyroid-stimulating hormone, TSH) production from the anterior pituitary gland. TRH is encoded by the prepro-TRH gene, which, through multiple steps of processing, generates the mature TRH which consists of 3 amino acid residues (pyro-Glu-His-Pro-NH$_2$). Prepro-TRH expression is negatively regulated by the liganded T3-receptor (TR) at the transcriptional level [5]. TRs are encoded by the TRα and TRβ genes [6]. TRα1 and TRα2 mRNAs are generated via alternative splicing of the TRα gene while usage of the different promoters in the TRβ gene generates TRβ1 and TRβ2 mRNAs. Among these TRs, T3 binds TRα1, TRβ1 and TRβ2 but not TRα2. TRβ2 expression is specific to the hypothalamus, pituitary, retina, and inner ear. Since repression of the prepro-TRH gene by T3 is ameliorated in TRβ2-knockout (KO) mice, TRβ2 is thought to mediate the negative regulation of this gene by T3 [7]. However, the mechanism by which the liganded TRβ2 inhibits prepro-TRH expression still remains unclear [1, 8].

In contrast, the molecular mechanism underlying the transcriptional activation of the T3 target gene (positive regulation) has been elucidated in detail [6]. In the genes that are positively regulated by T3, TRs heterodimerize with the retinoid X receptor (RXR) on a T3 responsive element (TRE). Typical TREs have a direct repeat 4 (DR4) structure, which consists of a tandem repeat of two short DNA sequences (AGGTCA), known as a halfsite, spaced by four nucleotides (spacer). The responsive elements for the vitamin D3 receptor (VDR) and the retinoic acid receptor (RAR) have a similar configuration except that the number of spacer nucleotides is 3 and 5, respectively (3-4-5 rule) [9]. In the absence of T3, the RXR-TR heterodimer recruits co-repressors, which associate with histone deacetylase, resulting in transcriptional repression. In the presence of T3, this heterodimer recruits the steroid receptor coactivator (SRC) family, which associates with the histone acetyltransferases (HATs) including the cAMP-responsive element binding protein-binding protein (CBP) or p300, resulting in transcriptional activation.

As mentioned above, TRH stimulates TSH expression of thyrotroph in the anterior pituitary [10]. TSH is a heterodimer with an α (chorionic gonadotropin α, CGA), and a β subunit (TSHβ). The latter determines the hormonal specificity of TSH. As in the case of prepro-TRH gene, both genes are negatively regulated by liganded TR [8]. For TSHβ genes, the presence of the negative TRE (nTRE) has been postulated as a counterpart of TREs in positively regulated genes [6]. Clinically, the relationship between positive and negative regulation appear to be a mirror image [11, 12]. If this is true at the molecular level, one may postulate that these genes may be activated by unliganded TR in an opposing manner to liganded TR. Based on this assumption, deletion study of the human TSHβ promoter was carried out with human kidney-derived 293 (HEK293) cells and an nTRE was reported as the DNA sequence which is necessary for activation in the absence of T3 [13]. This sequence, GGGTCA, is located immediately downstream of the transcriptional start site (TSS) and similar to the halfsite of TRE found in

positively regulated genes. In the TSHβ gene, this sequence has been believed to be the nTRE [14].

By analogy with the nTRE in the TSHβ gene, an nTRE was also postulated in the prepro-TRH gene. Using a firefly luciferase (FFL) gene fused to the human prepro-TRH promoter encompassing nt. -150 to nt. +55, Hollenberg et al. [15] reported that an inverted single halfsite (TGACCT) designated as site 4 may function as an nTRE. According to the authors, the mutation of it crippled the fold repression (FFL activity in the absence of T3 divided by that in the presence of T3) of this reporter gene. Site 4 is located at nt. -60 to nt. -55 in human (nt. -59 to nt. -52 in rat) and is highly conserved among species [16]. Although many investigators have accepted the physiological relevance of the site 4 [2–4, 17–19], the nTRE-based model raises some critical questions [8]. First, as Hollenberg described [1], it is difficult to explain how co-activators and/or co-repressors behave after T3 binding to TRβ2 on the nTRE. Second, in positive regulation, the number of the spacer nucleotides determines which nuclear hormone receptor (i.e. TR, VDR and RAR) dimerizes with RXR on the cognate responsive element [9]. However, the nTRE in the TSHβ genes [13] and site 4 [15] appear to be a single halfsite. If this is the case, the receptor specificity is difficult to explain [8]. Recent genome wide analysis rather suggested that the single halfsite may mediate positive regulation in combination with another halfsite located in a distal position [20]. Finally, it should be noted that the original report of the nTRE in the TSHβ promoter utilized HEK293 cells, and that there is no description of the use of expression plasmid of TR for mammalian cells [13]. However, subsequent studies report that endogenous TR is absent in these cells [21, 22]. Hollenberg et al. has suggested that it is necessary to be confirmed whether site 4 actually mediates the T3-dependent negative regulation in vivo. In subsequent study, they suggested the possibility that site 4 may function as a cAMP-responsive element (CRE) [23].

If the activity of the TSHβ promoter is maintained by unliganded TR, it should be reduced in TR-KO mice irrespective of T3 levels. However, TSHβ expression is up-regulated in these animals [24, 25]. Thus, unliganded TRs are not the activator of the TSHβ gene. This implies that (i), at the molecular level, T3-dependent negative regulation is not a mirror image of the positive regulation [26], (ii) the presence of the nTRE should be re-examined, and (iii) the transcription of the TSHβ in thyrotroph is maintained by factor(s) other than unliganded TRs [8]. GATA2 and Pit1 are the transcription factors that determine thyrotroph differentiation in the anterior pituitary and activate the TSHβ promoter [27]. We have found that inhibition by T3 is readily observed using the TSHβ promoter fused with chloramphenicol acetyltransferase (CAT) reporter gene in monkey kidney-derived CV1 cells [28] co-transfected with GATA2, Pit1, and TRs [26]. As predicted, unliganded TR failed to activate the TSHβ promoter in the absence of Pit1 and GATA2 [26]. Because CV1 cells have often been utilized in the studies of T3-dependent positive regulation [9], our findings suggested that pituitary-specific factor other than GATA2, Pit1, and TRs may not be necessary to mediate T3-dependent repression. We also found that the reported nTRE [13, 14] is dispensable for T3-dependent inhibition of the TSHβ promoter as long as these three transcription factors are co-expressed [29]. In solution and on the GATA-responsive elements (GATA-REs) of the TSHβ promoter DNA, we detected the protein-protein interaction of TRβ2 with GATA2 [29]. Taken together, we proposed that liganded TRs inhibit the TSHβ promoter via interaction between its DNA-binding domain and GATA2 (tethering), but not via direct binding with the DNA of the nTRE [8, 29].

Importantly, TRβ2-KO mice did not show a reduction in prepro-TRH expression either [7], suggesting that unliganded TRβ2 is not necessary for maintaining expression of the prepro-TRH gene. Therefore, the physiological relevance of site 4 should also be re-examined. As mentioned above, the rat prepro-TRH gene is known to be activated by the signals from multiple membrane receptors via cognate transcription factors including CREB, KLF10/TIEG1,

STAT3, SP-1 and AP-1 [1–4]. On the other hand, two the transcription factors, Sim1 and Arnt2, are reported to be essential for the differentiation of the PVN in the hypothalamus [30]. Interestingly, expression of GATA2 as well as TRβ2 is induced when Sim1 and Arnt2 are stably expressed in the neuronal cell line, Neuro-2a [31]. We speculated that, as in the case of TSHβ in the pituitary [29], TRβ2 and GATA2 may play roles in T3-dependent negative regulation of this gene in the PVN. Here, we report that (i) GATA2 protein is expressed in the TRH neuron of the rat PVN, (ii) a functional GATA-RE, dr-GATA, is located between nt. -357 and nt. -352 in the rat prepro-TRH promoter, (iii) liganded TRβ2 inhibits the GATA2-induced transcriptional activity of this gene, and (iv) when site 4 is mutated, T3-dependent negative regulation is maintained although the basal transcriptional activity is reduced.

## Materials and methods

### Animals

This study was in accordance with the Japanese Physiological Society's guidelines for animal care, and carried out in strict accordance with protocols approved by the Institutional Animal Care and Use Committee of Hamamatsu University School of Medicine, Hamamatsu, Japan (approval number; H28–053). Ten Wistar rats (six-week old male) were obtained from SLC Co. Ltd, Shizuoka, Japan. These rats had free access to food and water and were housed in temperature-(23 ± 2 C) and light (12-h light, 12-h dark cycle; lights on at 0700 h)-controlled conditions. Euthanasia was performed by overdose of sodium pentobarbital, and all efforts were made to minimize suffering.

### Immunohistochemistry tissue preparation

CV1 cells were plated on the culture cover (13mm, PLL-Cort C1100, Matsunami, Japan) and transfected with mouse GATA2 expression plasmid (pCDNA3-mGATA2) or empty vector using lipofectamine reagent. After washing with phosphate-buffered saline (PBS), cells were fixed with 4% paraformaldehyde (PFA)/PBS for 15 min at RT. Fixed cells were washed with PBS and permeabilized in 0.3% Triton-X 100/PBS for 5 min. Then, the cells were incubated with blocking solution containing 1% donkey serum in 0.1% Triton-X 100/PBS for 30 min at RT followed by incubation of 0.1% primary anti-GATA2 body (B9922A, Perseus Proteomics, Japan [32])/0.1% Triton-X 100/PBS overnight at 4˚C. After washing, the cells were incubated with 0.2% Alexa Fluor 488 anti-mouse IgG secondary antibodies (Thermo Fisher Scientific) for 30 min at RT followed by staining with 4',6-diamidino-2-phenylindole (DAPI). Images were acquired using a fluorescence microscope. Rats were anesthetized and intracardially perfused with 4% PFA/PBS for 5 min. Brains were dissected, post fixed with 4% PFA/PBS for 30 min, then incubated with 15% sucrose overnight. Subsequently, brain samples were incubated with 30% sucrose overnight at 4˚C. After immersion fixation, brains were frozen at -80˚C. Next, 20 μm cryostat coronal sections were washed with PBS. Endogenous peroxidase was quenched with 0.3% $H_2O_2$ in methanol for 20 min at RT. After washing with PBS, the sections were permeabilized in 0.3% Triton-X 100/PBS for 3 min. To enhance the antigen-antibody reaction, Histo VT One was used according to the manufacturer's instructions (Nacalai Tesque, INC, Japan). Next, the sections were incubated with blocking solution containing 3% bovine serum albumin (BSA) in 0.1% Triton-X 100/PBS for 30 min at RT followed by incubation with 0.2% anti-rabbit TRH primary antibodies (11170, Progen Biotechnic, Germany, a kind gift from Toshihiko Yada, Jichi Medical University, Japan)/0.1% Triton-X 100/PBS and 0.1% anti-mouse GATA2 primary antibodies (B9922A)/0.1% Triton-X 100/PBS overnight at 4˚C. The specificity of the anti-rabbit TRH primary antibodies was reported somewhere [33–35]. After the sections were incubated with horseradish peroxidase (HRP)-conjugated anti-

rabbit secondary antibody (DAKO, Denmark), they were wash with NTMT buffer (100 mM NaCl, 100 mM TrisHCl pH 9.5, 50 mM $MgCl_2$, 0.1% Tween20) and incubated with alkaline phosphatase (ALP)-conjugated anti-mouse secondary antibody (Jackson Immuno Research, PA, USA) for 30 min at RT. The areas treated with the HRP- and ALP-conjugated antibodies were visualized by ImmPACT DAB (DAKO, Denmark) and NBT/BCIP (Abcam, UK), respectively, according to the manufacturer's instructions.

## Plasmid constructions

The FFL-based reporter gene may be artificially suppressed by liganded TR [36–41]; therefore, we chose to use CAT-based reporter genes. The rat prepro-TRH promoter including nt. -547/+84 was fused to the CAT reporter gene, generating rTRH(547)-CAT. From the plasmid backbone in this construct, we deleted the pUC-derived AP-1-like sequence [26], which may also mediate the artifactual inhibition by liganded TR [42]. Deletion mutants, Del1, Del2, and Del3, were generated by polymerase chain reaction (PCR) with the following primers: 5′ primer for Del1 (5′–CATCTCTAGACTGCAGTCTGCCTTGCCCTCTCCC–3′), Del2 (5′–TGCTTCTAGACTGCAGCATCTGTCTTGTCTCTGG–3′), and Del3 (5′– GTTCTCTAGACTG CAGTTCTCTTAGTCAACAGACC–3′), and single common 3′ primer, (3′–AAAAAGATCTC GAGCAGAGCTTTCCAAGATGCTG–5′). PCR products were digested with Pst I and Bgl II and inserted into the rTRH(547)-CAT that had been digested with the same enzymes. Using site-directed mutagenesis, we mutated uf-GATG, dr-GATA, both uf-GATG plus dr-GATA, and site 4 in rTRH(547)-CAT to generate M1, M2, M3, and site 4m, respectively (Figs 3A and 4B). Expression plasmids for mouse GATA2 (pcDNA3-mGATA2) have been described previously [26]. We substituted the N-terminal domain of human TRβ1 (pCMX-hTRβ1) [9] with that of human TRβ2, to construct the expression plasmid for human TRβ2 (pCMX-hTRβ2). All subcloning sites and mutated sequences were confirmed by sequencing.

## Cell culture and transient transfection

CV1 cells ([28], the gift from Dr. Shunsuke Ishii (RIKEN Cluster for Pioneering Research, Tsukuba, Ibaraki, Japan) were grown in monolayer culture at 37˚C under $CO_2$/air (1:19) in Dulbecco's modified Eagle's medium (DMEM) containing 10% (v/v) fetal calf serum (FCS), penicillin G (100 units/mL), and streptomycin (100 mg/mL). Rat medullary thyroid carcinoma-derived CA77 cells (CRL-3234; [43]) were purchased from American Type Culture Collection (VA, USA) and maintained under the same conditions as CV1 cells except for the use of DMEM F-12 medium instead of DMEM. All cells were trypsinized and plated in 12-well plates for 16 h before transient transfection. When both cell lines reached 70% confluency per well, they were transfected using the Lipofectamine reagent according to the manufacturer's instructions (Promega, WI, USA). After cells had been exposed to Lipofectamine for 24 h, the medium was replaced with fresh media containing 5% (v/v) FCS depleted of thyroid hormone or media supplemented with T3 (0–100 nM) or 8-bromo cAMP (1 mM). After incubation for an additional 24 h, cells were harvested. CAT activities were normalized to β-galactosidase activities. For each reporter assay, we performed transfection with pCMV-CAT, the magnitudes of which were adjusted to a value of 100%. Mycoplasma testing with mycoplasma detection kit for conventional PCR, VenorGeM Classic (Minerva Biolabs Inc, Hillsborough, USA) showed negative for our CV1 cells and CA77 cells.

## Chromatin Immunoprecipitation (ChIP) assay

Approximately $1 \times 10^6$ CA77 cells were transfected with pcDNA3-mGATA2 and grown in 10 cm dishes. The cells were cross-linked with formaldehyde (1% final concentration) for 10 min

at RT. After cross-linking was terminated by the addition of glycine (0.125 M final concentration), cells were washed twice with ice-cold PBS, and collected by centrifugation. The cell pellets were resuspended in 200 μL of sodium dodecyl sulfate (SDS) lysis buffer (50 mM Tris-HCl, 10 mM EDTA, 1% SDS, 0.5 mM phenylmethyl sulfonyl fluoride, 2 μg/mL leupeptin, 2 μg/mL aprotinin), and incubated for 15 min on ice. Samples were sonicated 3 times for 10 s each and centrifuged at 14,000 rpm at 4˚C. The supernatants were diluted 10-fold with ChIP dilution buffer (50 mM Tris-HCl, 167 mM NaCl, 1.1% Triton X-100, 0.11% sodium deoxycholate (DOC)) supplemented with protease inhibitors. Chromatin solutions (2 mL) were pre-cleared with 60 μL of 50% protein G-Sepharose/salmon sperm DNA slurry (Upstate Biotechnology, Lake Placid, NY, USA), and incubated with 4 μL of antiserum against GATA2 (B9922A) overnight at 4˚C. Immunoprecipitated proteins were recovered with 20 μL of 50% protein G-Sepharose/salmon sperm DNA after 2 h and washed with low-salt buffer (50 mM Tris-HCl, 150 mM NaCl, 1 mM EDTA, 1% Triton X-100, 0.1% SDS, 0.1% DOC). Pellets were washed with high-salt buffer (50 mM Tris-HCl, 500 mM NaCl, 1 mM EDTA, 1% Triton X-100, 0.1% SDS, 0.1% DOC), followed by one wash with LiCl wash solution (10 mM Tris-HCl, 250 mM LiCl, 1 mM EDTA, 0.5% Nonidet P-40, 0.5% DOC), and two washes with Tris-EDTA. Protein-DNA complexes were eluted in the elution buffer (10 mM Tris-HCl, 300 mM NaCl, 5 mM EDTA, 0.5% SDS), and cross-linking was incubation at 65˚C for 4 h. The DNA was extracted using phenol-chloroform-isoamylalcohol (25:24:1) and precipitated with 20 μg of glycogen as a carrier. Samples were subsequently dissolved in 20 μL of TE. Using the SYBR Green I kit and Light Cycler (Roche Diagnostics, Mannheim, Germany), the ethanol-precipitated DNA was quantified by real-time PCR with primers designed to encompass dr-GATA in the rat TRH promoter (forward primer: 5′–GTGACACAGTCAAGCCCAGA–3′, reverse primer: 5′–GAGTAGTCCGCGATGGAAAG–3′). The thermal cycling conditions were as follows: 10 min at 95˚C, followed by 40 cycles of 10 s denaturing at 95˚C, 10 s annealing at 62˚C, and 7 s extension at 72˚C. PCR signals were analyzed using the Light Cycler software version 3.5 (Roche Diagnostics, Mannheim, Germany).

## Gel shift assay

Double stranded oligo DNAs for dr-GATA (probe drG, sense; 5′–AGATGCCACAAGTCCC TATCTCCTTTATTTTGCTGC–3′ and antisense; 5′–GCAGCAAAATAAAGGAGATA GGGACTTGTGGCATCT–3′, underline: GATA-RE) were labeled with $^{32}$P-ATP using a T4 polynucleotide kinase (Toyobo, Tokyo, Japan). CV1 cells were transfected with pcDNA3-m-GATA2 (15 μg per 10 cm dish). After incubation for 24 h, cells were harvested and nuclear extracts were prepared as previously described. The $^{32}$P-labeled probed dr-GATA and 2 μg of nuclear extracts were incubated for 30 min on ice in 20 μL binding buffer containing 10 mM Tris-HCl (pH 7.6), 50 mM KCl, 0.05 mM EDTA, 2.5 mM MgCl2, 8.5% (v/v) glycerol, 1 mM dithiothreitol, 0.5 μg/mL poly (dI-dC), 0.1% TritonX-100, and 1 mg/mL nonfat dry milk. DNA–protein complexes were resolved by electrophoresis on a 5% (w/v) polyacrylamide gel at 150 V for 180 min at 4˚C. For the supershift assay, antibodies against GATA2 (B9922A) were added to the binding reaction mixture. The gel was dried and labeled bands were visualized using the FLA-3000 autoradiography system (Fuji Film, Tokyo, Japan).

## Western blot analysis

Whole cell extracts of CA77 (299–598 μg/dish) cells or CV1 (114 μg/dish) cells transfected with human TRβ2 were fractionated (20 or 40 μL/lane) by SDS polyacrylamide gel electrophoresis (SDS-PAGE), and then, subjected to western blot analysis with an antibody that recognizes TRα, β1, and β2 (sc-32754, Santa Cruz Biotechnology Inc., Santa Cruz, CA).

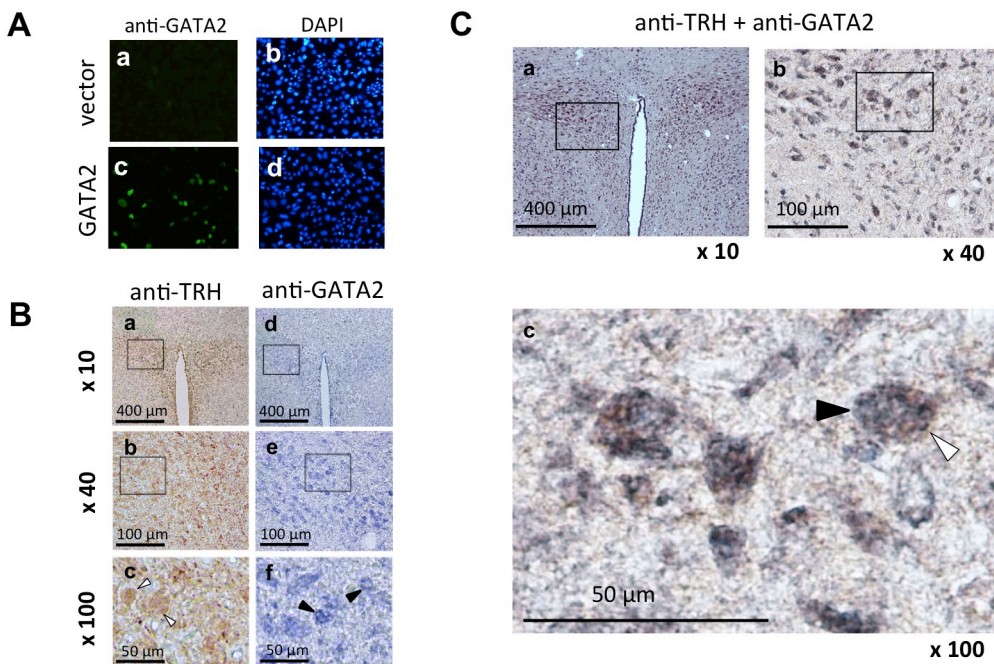

**Fig 1. GATA2 protein is expressed in the TRH neurons in the rat hypothalamic PVN.** (A) The specificity of anti-GATA2 antibody. CV1 cells were transfected with empty vector (a and b) or mouse GATA2-expression plasmid (c and d). These cells were stained with 0.1% anti-GATA2 antibody (B9922A, Perseus Proteomics, Japan) (a and c) or DAPI (b and d). Several GATA2-positive cells were detected in (c) but not in (a), while the numbers of nuclei stained with DAPI were comparable between (b) and (d). Transfection efficiency was approximately 5% (compare c and d). (B) (a, b, and c) Immunohistochemical staining of rat PVN with 0.2% anti-TRH antibody. Magnification: ×10 (a), ×40 (b), and ×100 (c). The cytoplasms were stained brown (open arrowhead). The boxed areas in (a) and (b) were magnified in (b) and (c), respectively. (d, e and f) Immunohistochemical staining of the rat PVN with anti-mouse 0.1% GATA2 antibody. Magnification: ×10 (d), ×40 (e) and ×100 (f). The nuclei were strongly stained blue (solid arrowhead). The boxed areas in (d) and (e) were magnified in (e) and (f), respectively. (C) Immunohistochemical double staining of the rat PVN with 0.2% anti-TRH antibody and 0.1% anti-mouse GATA2 antibody (a, b and c). Magnification: ×10 (a), ×40 (b), and ×100 (c). The cytoplasms of TRH neuron were stained brown with anti-TRH antibody (open arrowhead) and nuclei were stained blue with anti-mouse GATA2 antibody (solid arrowhead). The boxed areas in (a) and (b) were magnified in (b) and (c), respectively.

## Statistical analysis

Each CAT reporter assay was performed in duplicate three times, and each result was expressed as the mean ± standard error (S.E.). Significance was examined by analysis of variance (ANOVA) and Fisher's protected least significant difference test using EZR ver. 3.6.3 [44]. A value of $P < 0.05$ was considered to be statistically significant.

## Results

Sim1 and Arnt2 are the determinants for PVN differentiation in the hypothalamus [45, 46]. Previous studies have shown that stable expression of them induces TRβ2 and GATA2 expression in Nuero2a neuronal cells [31]. To examine GATA2 protein expression in the TRH neurons in rat PVN, we employed an anti-GATA2 antibody [10, 29, 32]. As shown in Fig 1A-a and 1A-c, the presence of GATA2 protein was detected by this antibody in the nuclei of CV1 cells transfected with GATA2 expression plasmid but not those with empty vector. The numbers of the cell nuclei were comparable between these dishes (Fig 1A-b and 1A-d). As shown in Fig 1B, frozen brain sections of adult male rats were immunohistochemically stained with

an anti-TRH primary antibody followed by an HRP-conjugated secondary antibody. This primary antibody is known to be able to detect the TRH expression in PVN [33, 34] and median eminence, where TRH is transported [35]. As shown in Fig 1B-a, 1B-b and 1B-c, we detected TRH-positive cells (brown in cytoplasms) in regions corresponding to PVN. Using sections of the PVN, we also observed numerous cells, nuclei of which were stained blue with the anti-GATA2 primary antibody and an ALP-conjugated secondary antibody (Fig 1B-d, 1B-e and 1B-f). Finally, double immunostain using anti-TRH and anti-GATA2 antibodies revealed that the nuclei of the TRH-positive neurons (brown in cytosol) were also GATA2-positive (blue in nuclei), suggesting the expression of GATA2 protein in TRH neurons (Fig 1C-a, 1C-b and 1C-c).

Balkan et al. [47] reported that the sequence from nt. -547 to nt. +84 of the rat prepro-TRH gene mediates the transcriptional activity in the hypothalamus while the sequence between nt. -776 and nt. -548 may have a silencing effect. Thus, we constructed rTRH(547)-CAT (Fig 2A) by fusing the sequence between nt. -547 and nt. +84 from the rat prepro-TRH promoter with the CAT reporter gene. As the pUC-derived AP-1 site may mediate the artifactual inhibition by liganded TR [42], this sequence was deleted from rTRH(547)-CAT. This reporter gene was

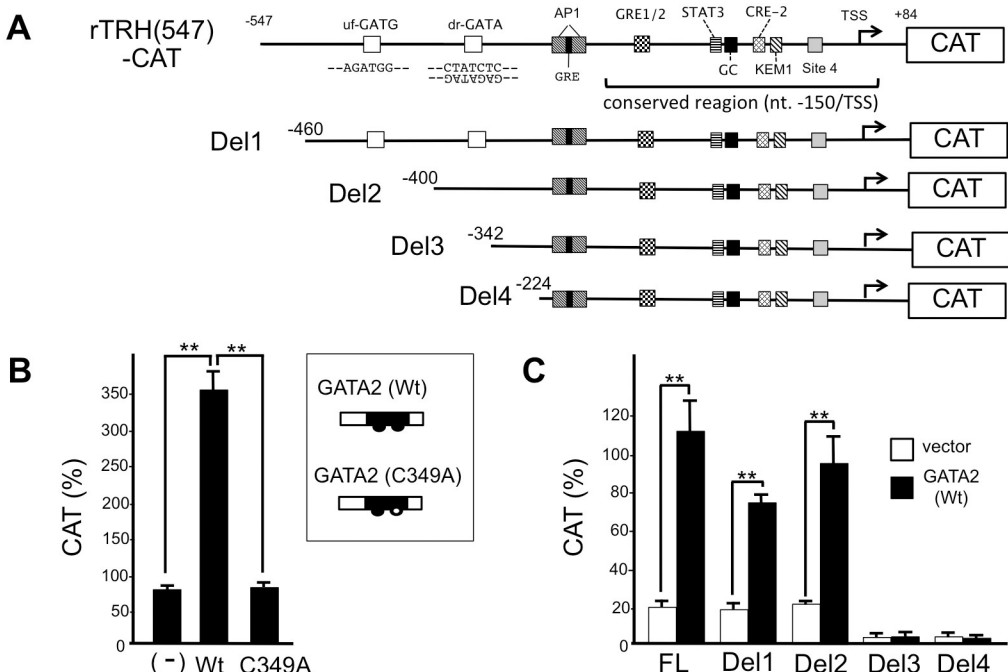

**Fig 2. The rat prepro-TRH gene is activated by GATA2.** (A) The rat prepro-TRH promoter (nt. -547 to nt. +84) was fused with CAT-reporter gene to generate rTRH(547)-CAT, from which the pUC-derived AP-1 site was deleted (see main text). Conserved region among species (nt. -150 to TSS) is indicated [16]. The candidate GATA sequences, upstream forward GATA (uf-GATG) and downstream reverse GATA (dr-GATA), are indicated. In addition to these GATA sequences, there are multiple short DNA elements that can be recognized by various transcription factors (see main text). Site 4; the reported negative T3 responsive element (nTRE) in prepro-TRH promoter. We also generated several deletion mutants (Del1, Del2, Del3 and Del4) of rTRH(547)-CAT. (B) In CV1 cells, CAT activities of rTRH (547)-CAT were enhanced by the transfection of mouse GATA2 expression plasmid (pcDNA3-mGATA2) but not by its mutant (C349A, inset). (C) GATA2-dependent transcription was abolished in the deletion construct (Del3 and Del4) suggesting the presence of functional GATA-RE(s) between nt. -400 and nt. -342. Using the Lipofectamine method, 2.0 μg of rTRH(547)-CAT (B) or its deletion mutants (C) were transfected into CV1 cells that were plated at a density of $2 \times 10^5$ cells per well in a six-well plate along with the expression plasmids for GATA2 or its mutant (C349A) (0.1 μg). **, $P < 0.01$ for vector vs. GATA2 expression plasmids. CAT activity for pCMV-CAT (5.0 ng/well) was taken as 100%. Data are expressed as the mean ± S.E. of three to five independent experiments.

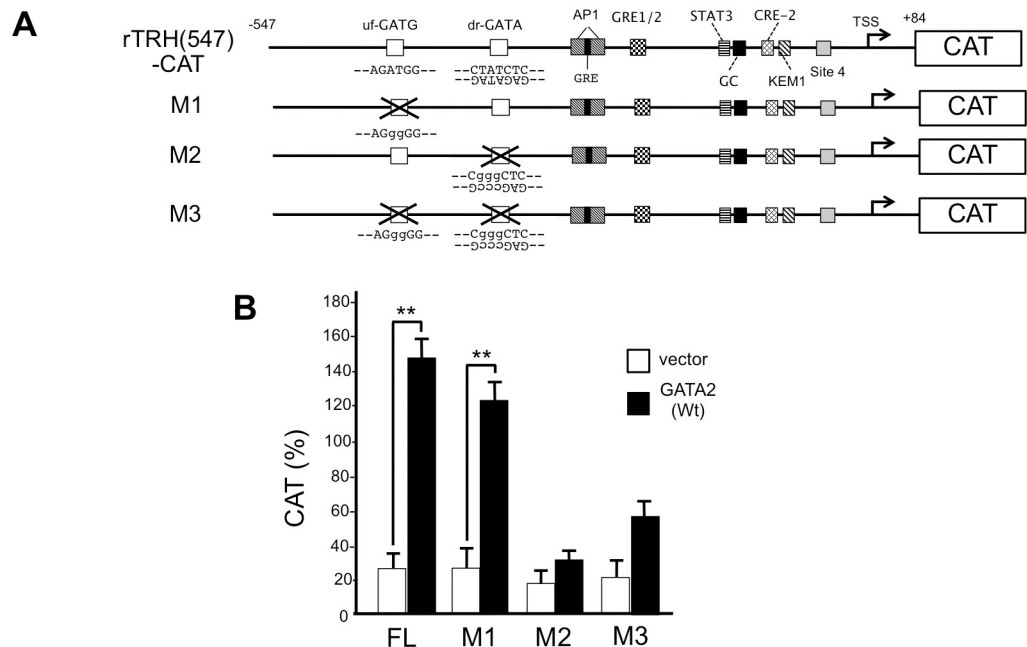

**Fig 3. In the rat prepro-TRH promoter, downstream reverse GATA sequence (dr-GATA) is the functional GATA-RE.** (A) Schematic representation of rTRH(547)-CAT and its mutant, M1, M2 and M3. (B) GATA2-induced transcription was significantly reduced by the mutation of dr-GATA (M2 and M3) but not uf-GATG (M1). Wild-type and mutated sequences are indicated as upper and lower case letters, respectively. Using the Lipofectamine method, 2.0 μg of rTRH(547)-CAT or its mutants (M1, M2, and M3) were transfected into CV1 cells that were plated at a density of $2 \times 10^5$ cells per well in a six-well plate along with GATA2 expression plasmid (0.1 μg). **, $P < 0.01$ for vector vs. GATA2 expression plasmids. CAT activity for pCMV-CAT (5.0 ng/well) was taken as 100%. Data are expressed as the mean ± S.E. of three to five independent experiments.

co-transfected into CV1 cells with a mouse GATA2 expression plasmid. As shown in Fig 2B, rTRH(547)-CAT was strongly activated by wild type mouse GATA2. We previously reported that a mutant GATA2, C349A, in which cysteine at the codon 349 in its DNA binding domain was substituted to alanine (Fig 2B, inset), failed to activate the TSHβ promoter [10, 29]. As expected, this mutant GATA2 again failed to potentiate the preproTRH promoter. These findings suggested the presence of a functional GATA-RE. Since GATA2 often recognizes redundant sequences that contain the sequence GAT [48, 49], we performed a computer search of the sequence between nt. -547 and nt. +84. We found two candidate sequences that may also be recognized by GATA2: the upstream sequence is in the forward direction (uf-GATG) while the downstream sequence is in reverse direction (dr-GATA). Our deletion analysis of rTRH (547)-CAT (Fig 2C) revealed that Del3 and Del4, but not Del1 and Del2 abolish promoter activity. We mutated uf-GATG and/or dr-GATA (Fig 3A) and observed that mutation of dr-GATA significantly reduced the transcriptional activity of rTRH(547)-CAT, suggesting that dr-GATA is the functional GATA-RE (Fig 3B). The modest activations of M2 and M3 constructs by co-transfection with GATA2 suggested the presence of cryptic weak GATA-RE(s).

As we have previously reported that liganded TRs interfere with the transcriptional activity of GATA2 in a T3-dependent manner [10, 26, 29], we tested the effect of T3 on the GATA2-induced promoter activity of preproTRH gene in the presence of TRβ2 (Fig 4A). We found that the activity of rTRH(547)-CAT was gradually reduced by addition of increasing concentrations of T3 (0–10 nM). This is consistent with our previous report that liganded TRs interfere with the transcriptional activity of GATA2 in a T3-dependent manner [10, 29, 49]. Next, we

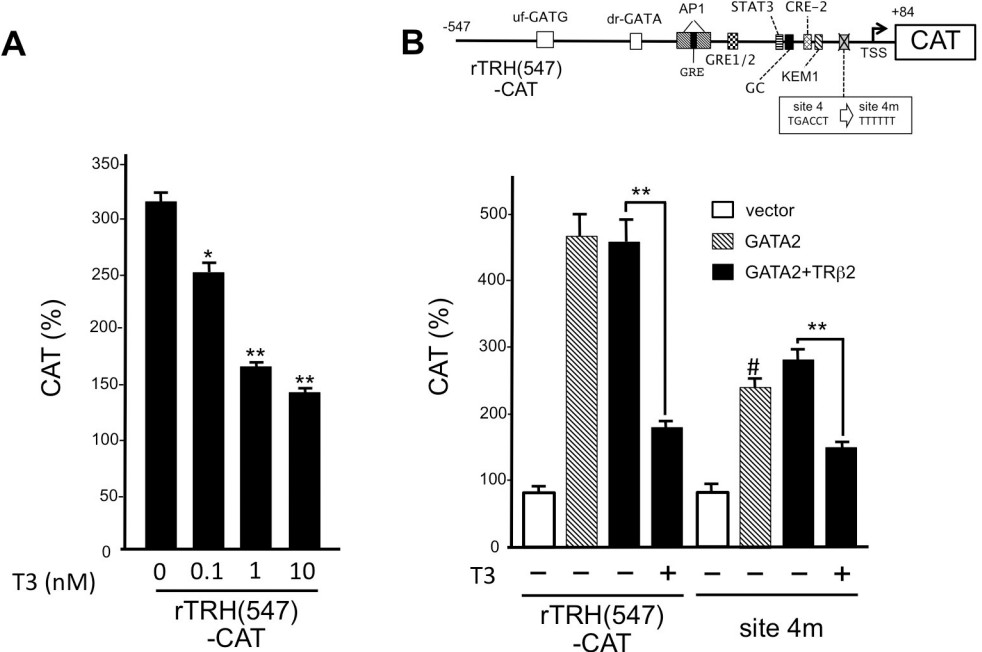

**Fig 4. The GATA2-dependent activation of the rat prepro-TRH promoter is inhibited by liganded TRβ2 and the reported nTRE, site 4, is dispensable for this inhibition.** (A) rTRH(547)-CAT was transfected into CV1 cells with GATA2 and TRβ2 expression plasmids and the cells were treated with 0 to 10 nM T3. The activity of this reporter gene was reduced by T3 treatment. *, $P<0.05$ vs. T3 (-) and **, $P<0.01$ vs. T3 (-) (B) Site 4 in rTRH(547)-CAT was mutated to generate site 4m. CV1 cells were transfected with this reporter gene and GATA2 and TRβ2 expression plasmids and the cells were treated 10 nM T3. In site 4m, the basal activity before T3 treatment is significantly reduced; however, T3-dependent negative regulation was maintained. **, $P<0.01$ vs. T3 (-) and #, $P<0.05$ vs. rTRH(547)-CAT.

examined the effects of mutating site 4 (site 4m, Fig 4B). Although the basal transcriptional activity of the prepro-TRH promoter was reduced by approximately 50% in site 4m, the T3-dependent inhibition of the GATA2-induced activity was maintained, suggesting that site 4 is dispensable for T3-dependent negative regulation of the rat prepro-TRH gene.

In the TRH neuron of the PVN, the α-melanocyte stimulating hormone (αMSH) neuron from hypothalamic arcuate nucleus and catecholamine neuron from the brainstem are known to stimulate the melanocortin 4 receptor and the catecholamine receptor, respectively [1, 3]. Both signaling events activate protein kinase A (PKA) via the CRE-binding protein (CREB), resulting in the activation of the prepro-TRH gene presumably via CRE1/2 and/or CRE2 (Fig 2A). Treatment with 1 mM 8-bromo-cAMP, an agonist of cAMP, displayed additive effects on the GATA2-dependent activity of rTRH(547)-CAT (Fig 5A). However, in the presence of GATA2, this activation was significantly inhibited by liganded TRβ2 (Fig 5B). These results indicate that the PKA signaling pathway may modulate the T3-dependent negative feedback regulation in the H-P-T axis.

A rat thyroid medullary carcinoma cell line, CA77 [43], is known to synthesize TRH [50–52]. According to Martinez-Armenta et al. [53], a kruppel-like factor (KLF)/Sp-1 family member, KLF10 (TIEG1), is expressed in CA77 cells and plays a crucial role in the expression of the rat preproTRH gene via its KLF-binding site (KEM1, Fig 2A). As we failed to detect the expression of endogenous GATA2, we transfected a GATA2 expression plasmid into CA77 cells and performed chromatin immunoprecipitation (ChIP) assays with an anti-GATA2 antibody. Using primers containing the dr-GATA sequence (Fig 6A), we detected *in vivo* binding of

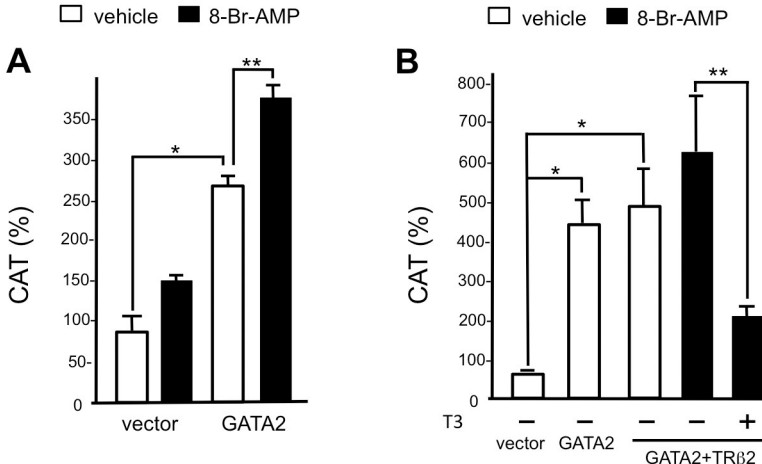

**Fig 5. PKA signaling pathway shows an additive effect on the T3-dependent negative regulation of the rat preproTRH gene by T3.** (A) Treatment with 1 mM 8-bromo-cAMP displayed an additive effect on the GATA2-dependent activity of rTRH(547)-CAT. rTRH(547)-CAT was transfected into CV1 cells with GATA2 expression plasmid in the presence or absence of 8-bromo-cAMP. **, $P<0.01$ vs. 8-bromo-cAMP (-). #, $P<0.05$ vs. vector. (B) GATA2-dependent activation in the presence of 8-bromo-cAMP (1 mM) was significantly inhibited by 10 nM T3 in the presence of TRβ2, indicating that the PKA signaling pathway may modulate the T3-dependent negative feedback regulation of the prepro-TRH gene. **, $P<0.01$ vs. T3 (-). #, $P<0.05$ vs. vector. The experimental procedures were the same as for Figs 2–4 but were conducted in the presence or absence of 8-bromo-cAMP and/or 10 nM T3.

GATA2 with this DNA element (Fig 6B). However, the binding signal was not statistically significant due to large standard deviations (p = 0.214 for control vs. dr-GATA transfected with GATA2 expression plasmid) presumably due to flexible conformational change of the GATA2 protein after binding with dr-GATA. We conducted gel-shift assays using $^{32}$P-radiolabeled double strand oligonucleotide encompassing the dr-GATA (probe drG) (Fig 6A) and the nuclear extract from CV1 cells transfected with GATA2 expression plasmid. As shown in Fig 6C, a single band was detected when GATA2 protein was incubated with $^{32}$P-radiolabeled probe drG. This signal was efficiently competed by a 50-fold excess of cold probe drG (lane 3) but not by its mutant (probe M. lane 4) or a nonspecific competitor (lane 5). In addition, this band was abolished by the preincubation of GATA2 protein with anti-GATA2 antibody (lane 6) as we previously shown with the GATA-REs in the TSHβ and D2 promoters [10, 49].

We detected endogenous expression of TRβ2 in whole cell extracts of CA77 cells by western blot with the antibody that recognizes TRβ2 (Fig 7A). As shown in Fig 7B, the activity of this promoter was potentiated by the transfection with GATA2 expression plasmid and it was negatively regulated by T3. T3 did not affect it when empty vector was transfected.

## Discussion

Sim1 and Arnt2 are two basic helix-loop-helix/PAS (Per-Arnt-Sim) transcription factors, which form heterodimers on the central nervous system midline enhancer (CME) to control the differentiation of neuroendocrine lineages in the hypothalamus [30]. Interestingly, SIM1--null mice show hypocellularity in PVN and supraoptic nucleus (SON) and absence of the expression of TRH, corticotropin-releasing hormone, oxytocin, vasopressin and somatostatin in these two nuclei and the anterior periventricular nucleus. Using Tet-On system in Neuro-2a cells, where Sim1 of which C-terminal domain was replaced with VP16 activation domain can be induced together with Arnt2, Liu et al. identified 268 potential target genes for Sim1/Arnt2

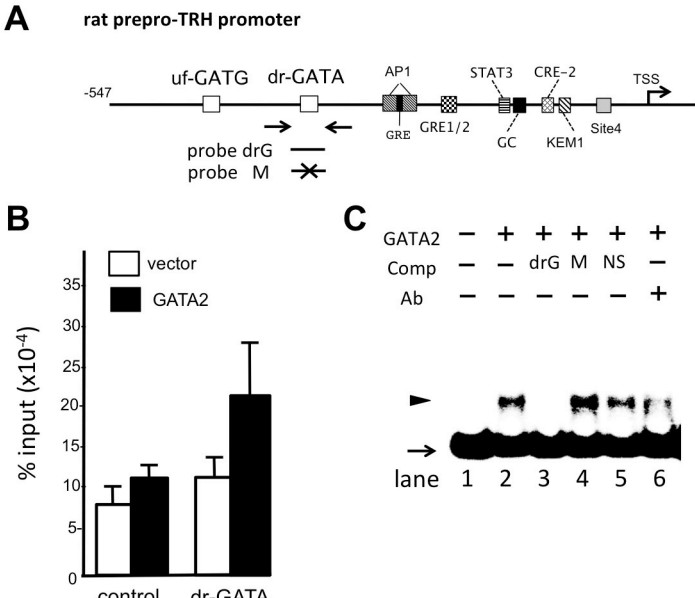

**Fig 6. GATA2 recognizes the dr-GATA sequence.** (A) Schematic representation of the rat prepro-TRH gene. The positions of primers for ChIP assay and the oligo-DNAs for gel shift assay (probe drG and its mutant probe M) are indicated. (B) A representative result of ChIP assays using an anti-GATA2 antibody and CA77 cells transfected with GATA2. Data are expressed as the mean ± S.E. of five independent experiments (p = 0.214 for control vs. dr-GATA transfected with GATA2 expression plasmid). (C) Gel shift assay with probe drG and its mutant probe M. A single band (arrowhead) was observed when $^{32}$P-radiolabeled probe drG was incubated with nuclear extract of CV1 cells transfected with mouse GATA2 expression plasmid. The specific binding signal (lane 2) was abolished by a 50-fold amount of cold probe drG (lane 3) but not by probe M (lane 4) or nonspecific double strand oligo-DNA (NS, lane 5). The signal was also reduced when GATA2 protein was mixed with the anti-GATA2 antibody before incubation with $^{32}$P-radiolabeled probe drG (supershift, lane 6). Arrow: free $^{32}$P-radiolabeled probe drG.

heterodimer [31]. As mentioned above, the GATA2 gene is included in addition to TRβ2 gene. According to Liu et al., there is a putative CME in the TRβ2 gene but not in the mouse pre-proTRH promoter [31]. Although the presence of CME in the GATA2 gene should be investigated in future, it should be noted that the regulatory mechanism of GATA2 gene in PVN may be different from that in thyrotroph because production of TSH was comparable to that of control pituitaries even in the absence of Sim1 [30].

Here, we confirm the presence of the GATA2 protein in the TRH neuron of the rat PVN (Fig 1). We also show that the T3-dependent negative regulation of the rat prepro-TRH gene is observed even in the CV1 cell as long as GATA2 and TRβ2 are co-expressed (Fig 8). Using primary culture cells from chicken hypothalamus, Lezoualc'h et al. [54] previously reported that T3 inhibits the activity of the CAT reporter gene fused to the rat prepro-TRH promoter between nt. -554 and nt. +84, a construct similar to our rTRH(547)-CAT; however, the details of its molecular mechanisms still remained unclear. Because CV1 cells lack endogenous TR [9] or GATA2 [29], our findings strongly suggest that liganded TRβ2 inhibits the GATA2-dependent transcription of the rat prepro-TRH gene presumably via a tethering mechanism as we previously proposed for the TSHβ gene [8, 29]. This hypothesis is also supported by other studies. First, the time course of the elevation of the rat prepro-TRH mRNA after the administration of an anti-thyroid drug, precisely correlates with that of TSHβ mRNA [55]. In thyroidectomized rats, the expression of these two genes is also elevated and the profiles of their down-regulation after T3 treatment are very similar [56]. Second, serum TSH is the most

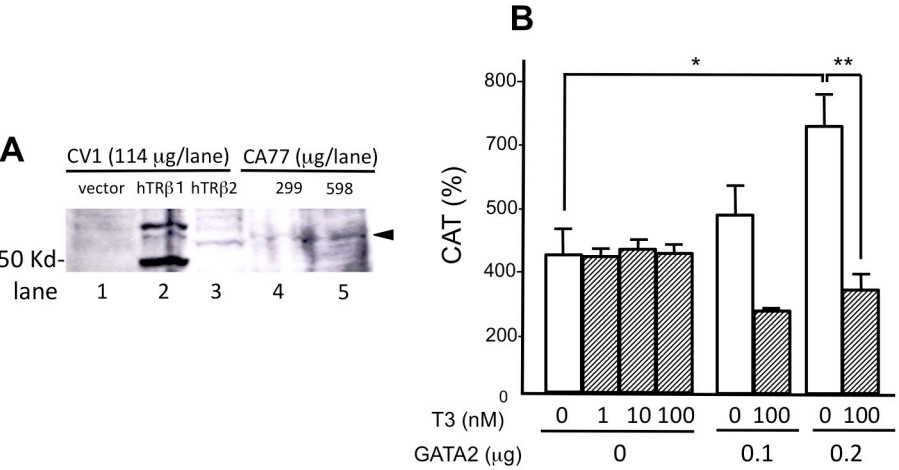

**Fig 7. T3-dependent negative regulation of the rat prepro-TRH promoter can be observed in rat thyroid medullary carcinoma cell line, CA77, transfected with a GATA2 expression plasmid.** (A) Endogenous TRβ2 in CA77 cells were detected by western blot of whole cell extract of CA77 cells using an anti-TR antibody. As a control, whole cell extract of CV1 cells transfected with empty vector (lane 1), human TRβ1 (hTRβ1, lane 2), or human TRβ2 (hTRβ2, lane 3) are shown. Endogenous rat TRβ2 (arrowhead) was detected by anti-TR antibody in 299 or 598 μg/lane of whole cell extract of CA77 cells. The difference of mobilities is thought to be the different molecular weights between human and rat TRβ2s. (B) rTRH(547)-CAT was transfected with a mouse GATA2 expression plasmid into CA77 ells. The additive effect by overexpression of GATA2 was observed and this activity is inhibited by T3, presumably via endogenous TRβ2. *, $P<0.05$ vs. vector. **, $P<0.01$ vs. T3 (-).

sensitive parameter for the function of the thyroid gland because it decreases exponentially after the linear increase of serum free T4 (log-linear relationship) [57]. Interestingly, the number of prepro-TRH positive cells in the rat PVN is also exponentially reduced after a linear increase of serum T3 concentration [58]. As the GATA2 gene harbors multiple GATA-REs and is enhanced by its own translation product, GATA2 protein, its expression is regulated via a non-linear positive feedback regulation [59]. We recently found that liganded TRβ2 represses this autoregulation, resulting in a drastic reduction of GATA2 transcription by T3 [60]. Third, estrogen moderately inhibits TSHβ transcription [14], as well as prepro-TRH gene expression [61]. These findings may be explained by our previous observation that liganded estrogen receptor α also partially interferes with the function of GATA2 via a tethering mechanism [62].

Here, we found that the negative regulation of the prepro-TRH gene is maintained even after mutation of site 4 when GATA2 and TRβ2 are co-expressed (Fig 4B). Originally, site 4 was reported by the analogy of the nTRE in the TSHβ gene, which was proposed based on the assumption that unliganded TR is a transcriptional activator for this gene [13]. However, as mentioned above, this assumption is not the case. Although the precise reason of the discrepancy between our study and the original report of site 4 [15] is unknown, it should be noted that the function of site 4 as an nTRE was determined using FFL-based reporter gene. When site 4 was reported [15], a paper suggested that FFL-based reporter gene may mediate the artificial negative regulation by T3 [36]. At that time, it was unclear if FFL cDNA per se may mediate it. Thus, the FFL assay has been utilized in many studies of the prepro-TRH gene after the report of site 4 [63–69]. However, subsequent analyses have suggested additional lines of evidence of this artifact [37–40], and Misawa et al. [41] finally demonstrated that FFL cDNA itself harbors a PKC-dependent enhancer sequence that is inhibited by liganded TR as in the case of the pUC-derived AP-1 site [42]. In the analyses of gene promoters, these artifacts may be negligible when the enhancer activities of the promoters under analyses are much higher than that

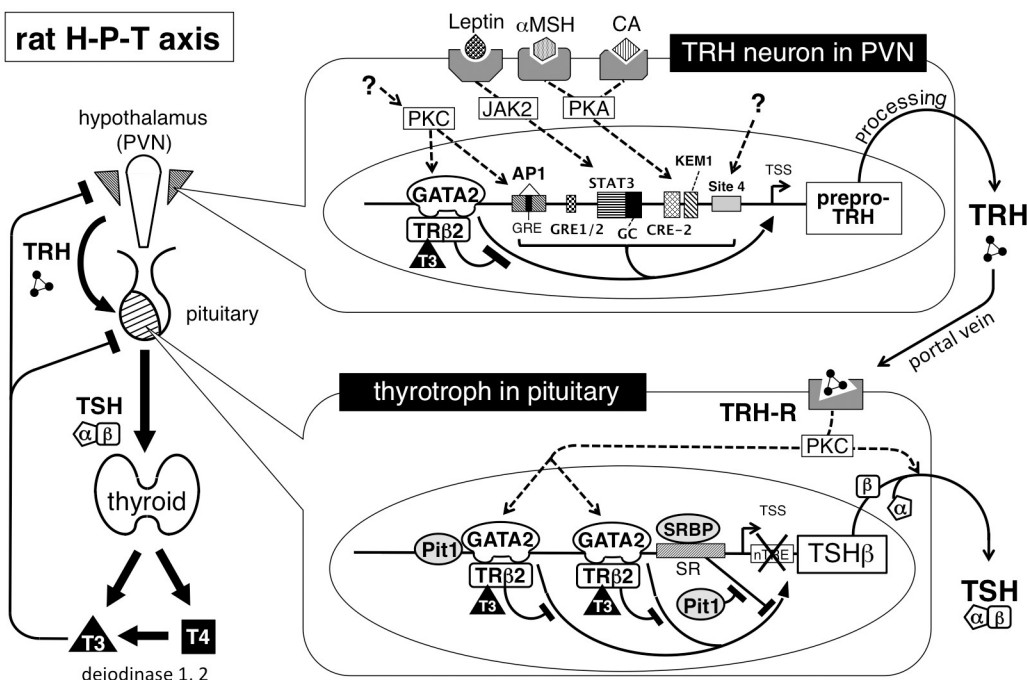

**Fig 8. A schematic representation of the transcriptional regulation of the rat prepro-TRH and TSHβ genes.** In the TRH neuron of the hypothalamic PVN (top) and thyrotroph of anterior pituitary lobe (bottom), TRβ2 associates with the Zn-finger region of GATA2 via protein-protein interactions, resulting in the T3-dependent interference of GATA2-dependent transactivation (tethering mechanism). The prepro-TRH gene is activated by multiple transcription factors stimulated by cognate cell membrane receptor signals (see main text), resulting in the fine-tuning of the set point of H-P-T axis. Broken lines indicate the signaling pathway via cell-membrane receptors/protein kinases. Although site 4 is dispensable for the T3-dependent negative regulation of the preproTRH gene, it may have physiological relevance at least in CV1 cells. In TRH neuron of the PVN, mature TRH (pyro-Glu-His-Pro-NH$_2$) is synthesized via multiple processing steps. In thyrotrophs, liganded TRH receptor stimulates the TSHβ and CGA gene expression and TSH secretion via the PKC pathway. In the TSHβ gene, Pit1 protects the transactivation function of GATA2 from the inhibitory effect by suppressor region-binding protein (SRBP) [8]. PKC, protein kinase C. JAK2, Janus kinase 2. PKA, protein kinase A. CA, catecholamine. αMSH, α-melanocyte stimulating hormone. GRE, glucocorticoid-responsive element. AP-1, AP-1 binding site. CRE, c-AMP-responsive element. TRH-R, TRH receptor. α, CGA. β, TSHβ. nTRE, the reported negative T3 responsive element in the TSHβ gene.

of FFL cDNA or the pUC-derived AP-1 site. However, genetic manipulation such as deletions or mutations often reduces their activities. Once the activities of the modified promoters decrease below that of FFL cDNA or the pUC-derived AP-1 site, the activities of modified reporter constructs may be artificially inhibited by liganded TR [8, 41]. Such confusion may occur unwittingly when the negative regulation by T3 is evaluated by the fold repression (FFL activity in the absence of T3 divided by that in the presence of T3). In our current and previous studies [10, 26, 29, 49, 70], we (i) mainly employed CAT- but not FFL-based reporter constructs, (ii) deleted the pUC-derived AP-1 site [42] from their plasmid backbone, (iii) analyzed the reported nTREs under the condition where basal transcription activity was maintained by GATA2 (Fig 4B), and (iv) did not depend on the evaluation with fold repression.

As shown in Figs 2A and 8, the GATA2-dependent transactivation of the prepro-TRH gene may be potentiated by the multiple transcription factors [3]. For example, catecholamine and αMSH signals induce phosphorylation of CREB via PKA pathway while leptin signal activates STAT3 via Janus kinase 2 pathway [1–4]. KLF10/TIEG1, which is expressed in CA77 cells and various parts of the adult rat brain, stimulates the rat prepro-TRH promoter via a KLF binding

site (KEM1, Fig 2A) [53]. In addition, prepro-TRH gene expression is also stimulated by protein kinase C (PKC) signaling via the AP-1 site [71] and, presumably, GATA-RE (dr-GATA) [10]. Although site 4 appears to be dispensable for T3-dependent inhibition, this sequence may be physiologically relevant [23] because its mutation (site 4m) decreases the basal promoter activity (Fig 4B). Thus, while the profiles of T3-dependent repression of them are very similar [55], the basal transcription level of the prepro-TRH gene is different from that of the TSHβ gene. This may explain the observation from 152,261 human subjects where the relationship between serum TSH and free T4 is not a simple log-linear curve [57], but is instead an overlap of two negative sigmoidal curves [72].

Using transgenic mice, Balkan et al. [47] reported that the DNA sequence between nt. -547 and nt. +84 of the rat prepro-TRH gene is required for the activity of this gene in hypothalamus. According to the authors, the transcriptional activity of the region encompassing nt. -243 to nt. +84 appears to be much higher in the olfactory bulb than that in the hypothalamus [47]. This is not the case in the normal rat brain, where the level of the prepro-TRH expression in the former is approximately half of that in the latter [73]. Although site 4 (nt. -59 to nt. -52) is included in this region, T3 does not inhibit the prepro-TRH expression in the rat olfactory bulb [73]. Thus, at least in rat, hypothalamus-specific expression and T3-dependent inhibition of this gene may require the region located upstream of nt. -243. This view is in agreement with our observation that the functional GATA-RE, dr-GATA, is located between nt. -357 to nt. -352 (Fig 2A). It should be noted, however, that only the DNA sequence between nt. -150 and TSS in the rat prepro-TRH promoter (Fig 2A) is conserved when it is compared with that of human [16]. Moreover, there are significant differences in the structures of the prepro-TRH genes among various vertebrates [16]. For example, the DNA sequences of the mammalian prepro-TRH genes have low homology with those of amphibians except for the short DNA sequences encoding TRH precursors. In frog, it is CRH but not TRH that mainly stimulates the expression of TSH from the anterior pituitary while TRH activates prolactin production, resulting in the regulation of osmolarity [74]. In some fishes, T3 does not affect the expression of TRH synthesis [75]. In this study, we chose the rat, but not mouse, preproTRH gene because (i) the anatomical and physiological findings of hypothalamus have been accumulated in this species, (ii) in vivo mapping of the rat preproTRH promoter was reported [47] and (iii) the transcription factors that regulate the rat preproTRH gene were characterized in detail (Figs 2 and 8). On the other hand, the screening of GATA-RE(s) in the prepro-TRH genes of the various species using expression plasmids for wild-type and mutant GATA2 (Fig 2B) may provide insights into the evolution of the H-P-T axis [16].

## Supporting information

**S1 File.**
(PDF)

**S1 Table.**
(XLSX)

**S1 Fig.**
(TIF)

**S2 Fig.**
(TIF)

**S3 Fig.**
(TIF)

**S4 Fig.**
(TIF)

**S5 Fig.**
(TIF)

**S6 Fig.**
(TIF)

**S7 Fig.**
(TIF)

**S8 Fig.**
(TIF)

**S9 Fig.**
(TIF)

**S10 Fig.**
(TIF)

**S11 Fig.**
(TIF)

## Acknowledgments

We would like to deeply thank Drs. Anthony N. Hollenberg (Weill Cornell Medicine, New York, New York, United States of America.) and Reiko Okada (Shizuoka University, Japan) for discussing this work with us. We are also grateful to Dr. Kazuhiko Umesono (Kyoto University, Japan) and Dr. Masayuki Yamamoto (Tohoku University, Japan) for providing plasmids and to Dr. Yasuharu Kanki (University of Tokyo, Japan) and Dr. Toshihiko Yada (Jichi Medical University, Japan) for providing the anti-GATA2 and anti-TRH antibodies, respectively.

## Author Contributions

**Conceptualization:** Shigekazu Sasaki.

**Data curation:** Go Kuroda, Shigekazu Sasaki, Akio Matsushita.

**Funding acquisition:** Shigekazu Sasaki, Takafumi Suda.

**Investigation:** Go Kuroda, Yuki Sakai, Shinsuke Shinkai, Hiroko Misawa Nakamura, Naoko Hirahara.

**Methodology:** Go Kuroda, Akio Matsushita, Kenji Ohba, Hiroko Misawa Nakamura, Satoru Yamagishi, Kohji Sato, Naoko Hirahara, Masahiko Ito, Tetsuro Suzuki.

**Project administration:** Shigekazu Sasaki.

**Software:** Kenji Ohba.

**Supervision:** Shigekazu Sasaki, Akio Matsushita, Yutaka Oki, Takafumi Suda.

**Writing – original draft:** Go Kuroda, Shigekazu Sasaki.

**Writing – review & editing:** Shigekazu Sasaki.

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
