## [Decision Letter · Decision Letter 0]

9 Sep 2020

PONE-D-20-24793

GATA2 mediates the negative regulation of the prepro-thyrotropin-releasing hormone gene by liganded T3 receptor β2 in the rat hypothalamic paraventricular nucleus

PLOS ONE

Dear Dr. Sasaki,

Thank you for submitting your manuscript to PLOS ONE. After careful consideration, we feel that it has merit but does not fully meet PLOS ONE’s publication criteria as it currently stands. Therefore, we invite you to submit a revised version of the manuscript that addresses the points raised during the review process.

We look forward to receiving your revised manuscript.

Kind regards,

Hiroyoshi Ariga

Academic Editor

PLOS ONE

Journal Requirements:

2. Please provide additional information about each of the cell lines used in this work, including source and any quality control testing procedures (authentication, characterisation, and mycoplasma testing). For more information, please see http://journals.plos.org/plosone/s/submission-guidelines#loc-cell-lines.

3. Please note that PLOS does not permit references to “data not shown.” Authors should provide the relevant data within the manuscript, the Supporting Information files, or in a public repository. If the data are not a core part of the research study being presented, we ask that authors remove any references to these data.

4. To comply with PLOS ONE submission guidelines, in your Methods section, please provide additional information regarding your statistical analyses, including the name and version of the specific software used in the analyses. For more information on PLOS ONE's expectations for statistical reporting, please see https://journals.plos.org/plosone/s/submission-guidelines.#loc-statistical-reporting.

5.We note that you have indicated that data from this study are available upon request. PLOS only allows data to be available upon request if there are legal or ethical restrictions on sharing data publicly. For more information on unacceptable data access restrictions, please see http://journals.plos.org/plosone/s/data-availability#loc-unacceptable-data-access-restrictions.

6.PLOS ONE now requires that authors provide the original uncropped and unadjusted images underlying all blot or gel results reported in a submission’s figures or Supporting Information files. This policy and the journal’s other requirements for blot/gel reporting and figure preparation are described in detail at https://journals.plos.org/plosone/s/figures#loc-blot-and-gel-reporting-requirements and https://journals.plos.org/plosone/s/figures#loc-preparing-figures-from-image-files. When you submit your revised manuscript, please ensure that your figures adhere fully to these guidelines and provide the original underlying images for all blot or gel data reported in your submission. See the following link for instructions on providing the original image data: https://journals.plos.org/plosone/s/figures#loc-original-images-for-blots-and-gels.

Reviewers' comments:

Reviewer's Responses to Questions

**Comments to the Author**

1. Is the manuscript technically sound, and do the data support the conclusions?

Reviewer #1: Yes

Reviewer #2: Partly

2. Has the statistical analysis been performed appropriately and rigorously? 

Reviewer #1: Yes

Reviewer #2: Yes

3. Have the authors made all data underlying the findings in their manuscript fully available?

Reviewer #1: Yes

Reviewer #2: No

4. Is the manuscript presented in an intelligible fashion and written in standard English?

Reviewer #1: Yes

Reviewer #2: Yes

5. Review Comments to the Author

Reviewer #1: Kuroda et al analyzed regulatory mechanisms of preproTRH promoter by thyroid hormone in rats. The authors demonstrated that GATA2, a key player of negative regulation by thyroid hormone, and TRH are coexpressed in the rat PVN. The authors also analyzed CA77 cells, which express TR beta 2 and TRH but not GATA2, and suggested downregulation of preproTRH gene expression is observed by supplementing the expression of GATA2. This is an interesting study and the authors have long series of publications exploring the mechanism of suppressed gene expression by thyroid hormone. This reviewer would like to point out following points that should be properly addressed by the authors.

Major points

Error bars and statistical analyses are required for the CHIP results in Fig. 6B.

If it is difficult, corresponding description in the main text should be modified.

Promoter activity in CA77 cells without GATA2 transfection should be presented in Fig. 7B.

It would be further nicer if the authors could demonstrate downregulation of endogenous preproTRH mRNA in CA77cells.

Minor

CA77 cell appears to be used, as it expresses preproTRH gene. This should be explained in the abstract.

Large parts of the introduction is devoted to describe historical changes in the concept of negative TREs, which has been described in series of publications by the authors. This reviewer feels that it should be shorter and conciser. On the other hand, molecular characteristics of Sim1 and Arnt2 should be further explained.

If the regulation of rat preproTRH promoter was analyzed for the first time by the authors in this manuscript, it should be more clearly described. The authors may discuss superiority of rats to mice as experimental animals in physiological studies.

Figure 4, uf-GATG is missing.

Figure 7A, whole cell extract should be indicated as mg protein rather than volume of the extracts. The amounts of protein applied in CV1 extracts should be also described.

Numbers of grammatical errors and unusual description are present. To attract broader readerships, English should be further edited. This is important as there is no chance to edit the accepted article in PlosOne. Followings are some of the examples of errors/unusual description.

Abstract

“found that this construct is activated by GATA2” sounds a bit odd. “showed that GATA2 activated the promoter” would be better.

Introduction

L1 Axis itself is not a mechanism. Axis may play the central role for maintaing…

L70 Various processing is a bit odd, sounds like various ways of processing. “Through multiple steps of processing” would be better.

L110 and thereafter, articles may be required before nTRE.

L113 and thereafter, site4 should be site 4.

L119 It is rare to see a word “himself” in scientific original articles.

L134 TSHß “promoter” or “gene”

L357, C349A should be first explained in the main text, not in the figure legends.

L420 Lezouaic ? or Lezoualc?

Reviewer #2: The authors have examined the role of GATA2 in the liganded TRbeta2 mediated negative regulation of pro-TRH gene. Although this manuscript is potentially interesting, I have the following concerns.

Major Points:

1) Immuno-staining is not convincing about the co-localization of GATA2 and TRH. The authors should perform double-staining of them using fluorescent antibody.

2) The authors did not show the direct interaction between GATA2 and TRbeta2. They should perform either immunoprecipitation or GST pull down assay.

Minor Points:

1) Recently, most people use luciferase assay instead of CAT assay. Why the authors used CAT assay in this manuscript?

6. PLOS authors have the option to publish the peer review history of their article (what does this mean?). If published, this will include your full peer review and any attached files.

Reviewer #1: No

Reviewer #2: No

---

## [Author Response · Author response to Decision Letter 0]

26 Oct 2020

REFERENCE: PONE-D-20-24793 

Title: " GATA2 mediates the negative regulation of the prepro-thyrotropin-releasing hormone gene by liganded T3 receptor β2 in the rat hypothalamic paraventricular nucleus”

AUTHORS: Go Kuroda et al. 

RESPONSE TO THE COMMENTS FROM REVIEWER

Reviewer #1

Major.

With regard to the error bars and statistical analyses for the ChIP results in Fig. 6B---

For more than 1 year, we have struggled with ChIP assay of GATA2 on the dr-GATA including the usage of the lentiviral vector-based induction of GATA2. However, we failed to determine its statistical significance. The data in Fig. 6B indicates the GATA2 binding with dr-GATA and we previously detected in vivo binding of it in our previous paper on the type 2 deiodinase gene (PLoS One. 2015; 10(11): e0142400.). We speculate that, because GATA2 protein can interact with DNA as a monomer, its conformation may be flexible, resulting in the alteration of epitope against antibody. Here, we modified the following sentence at the line 400 in the 1st submission. 

“However, the binding signal was not statistically significant due to large standard deviations (p=0.214 for control vs. dr-GATA transfected with GATA2 expression plasmid) presumably due to flexible conformational change of the GATA2 protein after binding with dr-GATA.” 

We added the standard error bar in the Fig. 6B and following sentence at the line 620 in the 1st submission. 

“Data are expressed as the mean ± S.E. of five independent experiments (p=0.214 for control vs. dr-GATA transfected with GATA2 expression plasmid). “

With regard to the promoter activity in CA77 cells without GATA2 transfection in Fig. 7B---

We conducted the reporter assay for the preproTRH promoter activity in CA77 cells without GATA2 transfection for three times and found that the activity was not affected by T3. We combined these data with previous one using the CAT activity of CMV-CAT as 100 (inter-assay control). We added the following sentence at the line 413 in the 1st submission. 

“T3 did not affect it when empty vector was transfected.”

With regard to the demonstration of downregulation of endogenous preproTRH mRNA in CA77cells---

We understand the importance of mRNA measuring. However, the repression by liganded TRß2 seems to depend on GATA2 (Fig. 7B). Because the transfection efficiency in CA77 is low (several %), more than 90% of CA77 cells do not express GATA2 after transfection. Thus, we speculate that the total amount of the preproTRH mRNA may not be affected by T3. 

Minor.

With regard to the expression of the preproTRH gene in CA77 in the abstract-----

We modified the following sentence at the line 53-57 in the1st submission. 

“A rat thyroid medullary carcinoma cell s, CA77, is known to express the preproTRH mRNA. Using a chromatin immunoprecipitation assay with this cell line where GATA2 expression plasmid was transfected, we observed the recognition of the GATA-RE by GATA2. We also confirmed GATA2 binding using gel shift assay with the probe for the GATA-RE.” 

With regard to the description of historical changes in the concept of negative TREs----

We agree with you now and deeply appreciate you following up our works. We modified the sentence at the lines 103-109 in the 1st submission as follows. 

Based on this assumption, deletion study of the human TSHß promoter was carried out with human kidney-derived 293 (HEK293) cells and a DNA sequence (GGGTCA) was reported as the nTRE which is necessary for activation in the absence of T3. 

We modified the following sentence at the lines 140-144 in the 1st submission as follows. 

We have found that inhibition by T3 is readily observed using the TSHß promoter fused with chloramphenicol acetyltransferase (CAT) reporter gene in kidney-derived CV1 cells [28] co-transfected with TRs in addition to GATA2 and Pit1 [26], two transcription factors that determine thyrotroph differentiation and activate the TSH� promoter. 

We modified the following sentence at the lines 151-154 in the 1st submission as follows. 

In solution and on the GATA-responsive elements (GATA-REs) of the TSHß promoter DNA, we detected the protein-protein interaction of TRß2 with GATA2 [29]. 

We deleted the sentences at the lines 158-160 and 447-452 in the 1st submission. 

With regard to the molecular characteristics of Sim1 and Arnt2---

We added a paragraph at the line 416 in the 1st submission 

Sim1 and Arnt2 are two basic helix-loop-helix/PAS (Per-Arnt-Sim) transcription factors, which form heterodimers on the central nervous system midline enhancer (CME) to control the differentiation of neuroendocrine lineages in the hypothalamus [30]. Interestingly, SIM1-null mice show hypocellularity in PVN and supraoptic nucleus (SON) and absence of the expression of TRH, corticotropin-releasing hormone, oxytocin, vasopressin and somatostatin in these two nuclei and the anterior periventricular nucleus. Using Tet-On system in Neuro-2a cells, where Sim1 of which C-terminal domain was replaced with VP16 activation domain can be induced together with Arnt2, Liu et al. identified 268 potential target genes for Sim1/Arnt2 heterodimer [31]. As mentioned above, the GATA2 gene is included in addition to TRß2 gene. According to Liu et al., there is a putative CME in the TRß2 gene but not in the mouse preproTRH promoter [31]. Although the presence of CME in the GATA2 gene should be investigated in future, it should be noted that the regulatory mechanism of GATA2 gene in PVN may be different from that in thyrotroph because production of TSH was comparable to that of control pituitaries even in the absence of Sim1 [30]. 

With regard to the mechanism of the regulation of the rat preproTRH promoter-----

We modified the following sentences in the line 164-165 in the 1st submission. 

As mentioned above, the rat prepro-TRH gene is known to be activated by the signals from multiple membrane receptors via cognate transcription factors including CREB, KLF10/TIEG1, STAT3, SP-1 and AP-1 [1-4]. On the other hand, two the transcription factors, Sim1 and Arnt2, are reported to be essential for the differentiation of the PVN in the hypothalamus [30]. 

We also modified the sentence at the lines 476-500 in the 1st submission as follows. 

As shown in Fig. 2A and 8, the GATA2-dependent transactivation of the prepro-TRH gene may be potentiated by the multiple transcription factors [3]. For example, catecholamine and αMSH signals induce phosphorylation of CREB via PKA pathway while leptin signal activates STAT3 via Janus kinase 2 pathway [1-4]. KLF10/TIEG1, which is expressed in CA77 cells and various parts of the adult rat brain, stimulates the rat prepro-TRH promoter via a KLF binding site (KEM1, Fig. 2A) [54]. In addition, prepro-TRH gene expression is also stimulated by protein kinase C (PKC) signaling via the AP-1 site [76] and, presumably, GATA-RE (dr-GATA) [10]. Although site 4 appears to be dispensable for T3-dependent inhibition, this sequence may be physiologically relevant [23] because its mutation (site 4m) decreases the basal promoter activity (Fig. 4B). Thus, while the profiles of T3-dependent repression of them are very similar [56], the basal transcription level of the prepro-TRH gene is different from that of the TSHß gene. This may explain the observation from 152,261 human subjects where the relationship between serum TSH and free T4 is not a simple log-linear curve [58], but is instead an overlap of two negative sigmoidal curves [73]. 

With regard to the superiority of rats to mice as experimental animals in physiological studies---

Because there are differences in the genomic structure of this gene among species, it is important to describe why we chose the preproTRH promoter of rats but not mice. We modified the sentence at the lines 521-523 in the 1st submission as follows. 

In this study, we chose the rat, but not mouse, preproTRH gene because (i) the anatomical and physiological findings of hypothalamus have been accumulated in this species, (ii) in vivo mapping of the rat preproTRH promoter was reported [47] and (iii) the transcription factors that regulate the rat preproTRH gene were characterized in detail (Fig. 2 and 8). On the other hand, the screening of GATA-RE(s) in the prepro-TRH genes of the various species using expression plasmids for wild-type and mutant GATA2 (Fig. 2B) may provide insights into the evolution of the H-P-T axis [16].

With regard to uf-GATG in Figures---

We added it in Figure 4 and 6. 

With regard to “mg” protein of CV1 whole cell extract---

We described the amount (mg) of whole cell extract of CV1 cell transfected with TRß2 expression plasmid and that of CA77 cells at the lines 317 and 635 in the1st submission. 

We also added “mg” protein of CV1 whole cell extract in the section of Materials and Methods and Fig. 7A. 

With regard to the grammatical errors and unusual description---

We corrected all the words and the sentences that you kindly pointed. 

With regard to mutant GATA2 (C349A) ---

We modified the sentences in the line 356-359 as follows.

“As shown in Fig. 2B, rTRH(547)-CAT was strongly activated by wild type mouse GATA2. We previously reported that a mutant GATA2, C349A, in which cysteine at the codon 349 in its DNA binding domain was substituted to alanine (Fig 2B, inset), failed to activate the TSHß promoter [10, 29]. As expected, this mutant GATA2 again failed to potentiate the preproTRH promoter, suggesting the presence of a functional GATA-RE.”

In addition, we deleted the sentence “in which cysteine at the codon 349 in its DNA binding domain was substituted to alanine” at the line 589-590 in the 1st submission. 

With regard to Lezouaic'h in the line 420 in the 1st submission---

We corrected it as Lezoualc'h. 

Reviewer #2: 

With regard to double-staining of TRH neurons using fluorescent antibody----

We have attempted to do double-staining of them using fluorescent antibody for more than 8 months. However, we failed to detect signals with anti-GATA2 (BA9922A, Perseus Proteomix, Japan) although we were able to detect the signal of immune-fluorescent stain using cultured cells (CV1, Fig. 1A). To our knowledge, there is no report of the immuno-fluorescent stain of tissue using this anti-body. Thus, we speculate that this phenomenon is the characteristics of this antibody. Although we attempted again immuno-fluorescent stain of PVN using another anti-GATA2 antibody (H-116, Santa Cruz Biotechnology, USA) three times for the revised paper, we again failed. However, we believe the specificity of our anti-GATA2 antibody (BA9922A, Figure 1A) and the co-localization of GATA2 protein at least in some population of TRH neurons (Figure 1C-c). 

With regard to co-immunoprecipitation or GST pull down assay between GATA2 and TRß2---

As described in the line 154-155 in the 1st submission, we already reported both co-immunoprecipitation and detailed GST pull down assay between GATA2 and TRß2. Please see S10 Fig.pdf and S11 Fig.pdf in the “Supporting information3 (a ZIP file)” which were published from our laboratory (reference [29], Mol Endocrinol. 2007 (4):865-84, https://academic.oup.com/mend/article/21/4/865/2738390). In addition, we modified the following sentence at the lines 151-154 in the 1st submission as follows. 

In solution and on the GATA-responsive elements (GATA-REs) of the TSHß promoter DNA, we detected the protein-protein interaction of TRß2 with GATA2 [29]. 

With regard to the reason why we used CAT but not luciferase assay in this manuscript---

As mentioned in the line 227-228 in the 1st submission (the section of “Materials and Methods”) and reference [41], we previously evaluated the enhancer activity of firefly luciferase (FFL) cDNA using CAT-based assay and found that some DNA sequence of it may function as an enhancer that can be stimulated by PKC and repressed by T3-bound TR. (PLoS One. 2012 (1):e28916, https://journals.plos.org/plosone/article?id=10.1371/journal.pone.0028916). We attached these data as S12 FigLucCAT1 and S13 FigLucCAT2 in the “Supporting Information3 (a ZIP file)”. Although the artifact with FFL cDNA may be unexpected, it has profoundly impeded the studies on the T3-dependent negative regulation, in particular, of the preproTRH gene. Please see also the description in the line 452-475 and references in the 1st submission.

---

## [Decision Letter · Decision Letter 1]

2 Nov 2020

GATA2 mediates the negative regulation of the prepro-thyrotropin-releasing hormone gene by liganded T3 receptor β2 in the rat hypothalamic paraventricular nucleus

PONE-D-20-24793R1

Dear Dr. Sasaki,

We’re pleased to inform you that your manuscript has been judged scientifically suitable for publication and will be formally accepted for publication once it meets all outstanding technical requirements.

Kind regards,

Hiroyoshi Ariga

Academic Editor

PLOS ONE

Reviewers' comments:

Reviewer's Responses to Questions

**Comments to the Author**

1. If the authors have adequately addressed your comments raised in a previous round of review and you feel that this manuscript is now acceptable for publication, you may indicate that here to bypass the “Comments to the Author” section, enter your conflict of interest statement in the “Confidential to Editor” section, and submit your "Accept" recommendation.

Reviewer #1: All comments have been addressed

Reviewer #2: All comments have been addressed

2. Is the manuscript technically sound, and do the data support the conclusions?

Reviewer #1: (No Response)

Reviewer #2: Yes

3. Has the statistical analysis been performed appropriately and rigorously? 

Reviewer #1: (No Response)

Reviewer #2: Yes

4. Have the authors made all data underlying the findings in their manuscript fully available?

Reviewer #1: (No Response)

Reviewer #2: Yes

5. Is the manuscript presented in an intelligible fashion and written in standard English?

Reviewer #1: (No Response)

Reviewer #2: Yes

6. Review Comments to the Author

Reviewer #1: (No Response)

Reviewer #2: The authors have adequately addressed to my comments including protein-protein interaction and Luc assay.

7. PLOS authors have the option to publish the peer review history of their article (what does this mean?). If published, this will include your full peer review and any attached files.

Reviewer #1: No

Reviewer #2: No

---

## [Editor Report · Acceptance letter]

9 Nov 2020

PONE-D-20-24793R1 

G ATA2 mediates the negative regulation of the prepro-thyrotropin-releasing hormone gene by liganded T3 receptor β2 in the rat
hypothalamic paraventricular nucleus 

Dear Dr. Sasaki:

I'm pleased to inform you that your manuscript has been deemed suitable for publication in PLOS ONE. Congratulations! Your manuscript is now with our production department. 

Kind regards, 

on behalf of

Dr. Hiroyoshi Ariga 

Academic Editor

PLOS ONE